# Obtaining Bioproducts from the Studies of Signals and Interactions between Microalgae and Bacteria

**DOI:** 10.3390/microorganisms10102029

**Published:** 2022-10-14

**Authors:** Évellin do Espirito Santo, Marina Ishii, Uelinton Manoel Pinto, Marcelo Chuei Matsudo, João Carlos Monteiro de Carvalho

**Affiliations:** 1Department of Biochemical and Pharmaceutical Technology, Faculty of Pharmaceutical Sciences, University of São Paulo, São Paulo 05508900, Brazil; 2Research Center, Department of Food and Experimental Nutrition, Faculty of Pharmaceutical Sciences, University of São Paulo, São Paulo 05508900, Brazil; 3Center for Natural and Human Sciences, Federal University of ABC, Santo André 09210580, Brazil

**Keywords:** quorum sensing, co-culture, biomass, phytoplankton

## Abstract

The applications of microalgae biomass have been widely studied worldwide. The classical processes used in outdoor cultivations of microalgae, in closed or open photobioreactors, occur in the presence of bacteria. Understanding how communication between cells occurs through quorum sensing and evaluating co-cultures allows the production of microalgae and cyanobacteria to be positively impacted by bacteria, in order to guarantee safety and profitability in the production process. In addition, the definition of the effects that occur during an interaction, promotes insights to improve the production of biomolecules, and to develop innovative products. This review presents the interactions between microalgae and bacteria, including compounds exchanges and communication, and addresses the development of new pharmaceutical, cosmetic and food bioproducts from microalgae based on these evaluations, such as prebiotics, vegan skincare products, antimicrobial compounds, and culture media with animal free protein for producing vaccines and other biopharmaceutical products. The use of microalgae as raw biomass or in biotechnological platforms is in line with the fulfillment of the 2030 Agenda related to the Sustainable Development Goals (SDGs).

## 1. Introduction

The annual production of microalgae is estimated to be up to 19,000 tons per year [1], with great commercial potential. Data Bridge Market Research estimated the growth of the microalgae market from 2021 to 2028, concluding that it may reach US$ 62 billion in 2028. This demand is due to the increasing consumer awareness regarding the health benefits of products of microalgal biomass as well as the high demand for proteins of non-animal origin [2]. According to Transparency Market Research, there will be a faster expansion of the microalgae market in North America, followed by Europe and Asia-Pacific. The global market for microalgae-derived products is estimated at $4.2 billion in 2031, in terms of revenue expansion at a Compound Annual Growth Rate (CAGR) of approximately 6% [3]. Phytoplankton biomass is fundamental for the green economy, providing more food resources without the need for a greater territorial extension for agriculture, and helping to solve the dilemmas of fossil fuel depletion [4]. The use of renewable resources is in line with the Sustainable Development Goals (SDGs) of the 2030 Agenda, and microalgae biomass has gained importance but is still little explored in this context [5].

Considering the biocompounds produced by microalgae, carotenoids and polyunsaturated fatty acids have high added value, with an essential demand for naturally synthesized carotenoids, in global markets [5]. The most cultivated photosynthetic microbial species for obtaining biomass and extracting bioproducts are those of the genera *Dunaliella*, mainly for obtaining β-carotene, *Haematococcus*, as a source of astaxanthin, as well as *Chlorella* and *Arthrospira* used as food additives [6]. In addition to the carotenoids produced by biopharmaceutical industries, these microalgae are essential sources of potential molecules for various cosmetics, food, and animal feed [7]. Although, currently, the use of microalgae has been competitive in the market only when all fractions are valued [5], efforts are needed to improve the production process due to the fact that production of microalgae and cyanobacteria can be negatively impacted by some species of bacteria, fungi, and viruses [8]. Therefore better understanding of the interactions between algae and other organisms must be considered to grant safety and productivity in production process. Such a guarantee is crucial for producing any industrial product, assuring confidence for the investors and producers. On the other hand, other species may have a positive impact, as it takes place in nature [9].

There is little information available on microalgae bioproducts and their applications in the literature, mainly referring to uses in nutraceutical, cosmetical, pharmaceutical, and the food industry [10]. Biological products are on the rise in industries, and their development is essential for valorizing materials from biomass. It is necessary to find alternatives to improve biomolecules production and apply them. For example, despite the promising applications of antibacterial peptides and the importance of searching for new natural sources of antibiotics, limitations to their pharmaceutical applications persist [11].

It is essential to evaluate microalgae-bacteria interactions to foresee the behavior of the consortia system, allowing adjustments in the operational conditions for successful microalgae production. Existing co-culture articles focus on the cultivation process, especially algal growth [7,9,12], and synthetic ecology for consortium assembly to be more productive and/or more resistant to contamination [13,14]. They provide excellent insights into wastewater treatment [15,16], removal of pollutants while producing metabolites of interest [17], bioremediation [16], aquatic ecosystems [16,18], and biofuels [19]. This review focuses on the microbial cultivation process applied to the production of high- value-added biomolecules in biopharmaceutical, cosmetics, and food products. The success in cultivating microalgae under heterotrophic conditions is recognized [20], but this type of cultivation has the disadvantage of the need for axenic culture, and any contamination by heterotrophic microorganisms is strictly prohibited, as they may grow faster than microalgae. From the productive point of view, we aim to increase the cell concentration and these biomolecules, within stable, safe and robust processes, in order to guarantee the quality and profitability of the production under photoautotrophic conditions. It also presents an analysis of the interaction mechanisms between microalgae and bacteria, including quorum sensing evaluation, inhibitory and beneficial effects of metabolites, direct interaction and co-immobilization in co-cultures in order to use this knowledge to develop new bioproducts employing microalgae, such as prebiotics, antibiotics, and vegan culture media, contemplating applications in medicines, cosmetics, and food.

## 2. Microalgae-Bacteria Interaction

In 1958, an Italian researcher named Luigi Provasoli suggested that bacteria could improve algae growth [21]. It is known that bacterial communities live together with microalgae and can conduct their metabolism in consortia systems [15,17]. In nature, interactions between bacteria and microalgae play an indispensable role in maintaining the integrity of the aquatic ecosystem through networks of interactions such as competition and mutualism [22]. In fact, in the wild, the growth of algae is consistently associated with the growth of other microorganisms, especially bacteria. Axenic culture systems and sterilization of culture media in large-scale production of microalga are not economically feasible. Therefore, the characterization of associated heterotrophs in algae culture systems is an important step since bacteria may use compounds excreted by algae, increasing the availability of trace elements and solubility of nutrients, making them more bioavailable for microalgae. In addition, they can help to reduce the saturation of dissolved O_2_ [23]. In microalgae cultivation, it is well known that dissolved O_2_ can attain inhibitory levels [24].

For the production of microalgae in non-axenic conditions, it is necessary to ensure that the bacterium, which is present in the cultivation of the microalgae, does not present any toxicity to the latter. In addition, the concentration of bacteria must be maintained below a maximum threshold, promoting microalgae growth and preventing competition for nutrients [25]. The environmental conditions determine the growth and death of these microorganisms. Since microalgae’s death releases several organic compounds in the cultivation medium, cultivation in optimum conditions avoids cell death and, consequently, excessive bacterial growth. On the other hand, microalga can also harm bacterial metabolism by producing antibacterial compounds, increasing the concentration of dissolved oxygen in the cultivation medium, or increasing the pH.

The environmental conditions also determine the species present in the cultivation and, consequently, the type of interaction between microorganisms. For example, microalgae growth in extreme salinity avoids competition and contamination with less halotolerant organisms [23,26].

It is known that the physiological changes mediated by bacteria and microalgae include increases in algae growth, flocculation, and a change in lipid content [22]. Besides, these interactions can even contribute to avoiding contamination of the culture with other microorganisms. On the other hand, it was observed that after antibiotic treatment, a bacteria-free *Dunaliella salina* culture was obtained, but fungal contaminants were observed [26].

The knowledge about possible interactions between bacteria and algae allows the control of the mechanisms that support mutualistic interactions, which can improve the development of bioprocesses with increased algae biomass and production of biocompounds such as carotenoids, lipids, and carbohydrates with high added value [22]. Also, this knowledge leads to the development of possible microalgae-based commercial products, including food, cosmetic and pharmaceutical products. Figure 1 shows possible interactions between microalgae and bacteria.

### 2.1. Microbial Interactions through Quorum Sensing

Quorum sensing (QS) is a phenomenon of communication between microorganisms through the release and detection of small signaling molecules that accumulate in their surroundings as a function of cell-density [27]. Cell-to-cell communication system is used by bacteria in order to assimilate environmental cues and to monitor microbial population density, ultimately affecting gene expression and microbial group behavior responses [28].

There are several QS communication systems, and the most well-known mechanisms are described in bacteria. QS is usually based on the production, secretion and detection of low molecular weight signaling biomolecules commonly referred to as autoinducers (AI). Quorum sensing plays essential roles in regulating gene expression and modulating complex processes such as virulence, symbiosis, production of bacteriocins, sporulation, conjugation, bioluminescence, motility, and biofilm formation, among others [29].

Signaling molecules vary according to each bacterial group. In Proteobacteria, AI molecules belong to the group of acylated homoserine lactones (AHLs), initially known as autoinducer-1 (AI-1). AHL molecules vary in size and substitutions in the carbon chain, creating specificity in the communication within the species level. However, it is known that AHL molecules from one species can be detected by homologous signaling receptors present across different bacterial species [30]. Autoinducer peptides (AIP) are used by Gram-positives as QS mediators, presenting significant structural variations among different organisms. A molecule known as autoinducer 2 (AI-2), derived from 4,5-Dihydroxy-2,3-pentanedione (DPD), is produced universally and has been associated with interspecific bacterial communication. Additionally, many other small signaling molecules belonging to different chemical classes, mediate cell-to-cell communication within different microbial groups, including bacteria, yeast, and fungi. Quorum sensing signaling is prevalent in the microbial world with a diverse signaling repertoire and a range of cell-density population responses [29].

Microbial communication through signaling molecules has been intensively investigated, and two-way interactions between bacteria and other organisms have been reported across different fields of study. For instance, Kim and collaborators (2020) described a type of interkingdom communication between *Escherichia coli* and humans through the host-derived adrenergic signals norepinephrine and epinephrine that activate bacterial responses, meanwhile, the bacterial signaling molecules collectively known as autoinducer 3 (AI-3) have been shown to exert immunological effects in human tissues [31]. Studies concerning interactions of microorganisms in the rhizosphere (the zone surrounding the roots of plants) have revealed that plant species may respond to bacterial autoinducers by secreting phenolic compounds that act as inhibiting contaminants, in addition to interfering in their signaling pathways [32]. Similarly, phytoplankton-bacteria interactions have also been investigated, and studies have shown that besides the exchange of nutrients, chemical signals are also important players in the phycosphere, which is the region close to the microalgal cell where bacteria may thrive [33,34]. Some microalgae can synthesize quorum sensing mimics that can affect bacterial communication and behavior [35].

Fei and collaborators (2020) have shown that QS signals enabled the colonization of the phycosphere of the diatom *Asterionellopsis glacialis* by symbiotic bacteria of the *Roseobacter* group through inhibition of motility and improvement of biofilm formation, which likely enhanced the capacity of the bacteria to attach to the diatom microalgae. Interestingly, diatoms may also interfere with bacterial motility by producing QS mimics, such as rosmarinic acid, which may contribute to bacterial attachment [36]. Seymour et al. (2017), pointed out that the phycosphere may represent a rich site for chemical signal exchanges. In fact, the interactions between phytoplankton and bacteria seem to be highly sophisticated, and these interactions influence many processes that can affect the system’s productivity, either positively or negatively [18].

Quorum sensing molecules (QSMs) extracted from wastewater microbial consortiums increased the production of lipid content by the microalgae *Chlorophyta* sp. by 86%, but algal biomass was slightly reduced [37]. The authors hypothesized that QSMs triggered an environmental pressure on the microalgae, having a positive effect on microalgae biofuel production. In another study, when adding bacterial QSMs extracted from anaerobic bacterial sludge of a microbial fuel cell to a photobioreactor containing the microalgae *Chlorella sorokiniana*, there was an improvement in algae biomass productivity by 2.25 times, as well as significant increases in the lipid and protein content, as well as in the harvest efficiency [38]. The supposedly QSMs present in the extract included several N-acyl homoserine lactones, bacterial siderophores, oligopeptides, and vitamin B12. Thus, it seems that besides true QS signaling molecules represented by the AHLs detected in the extracts, cross-feeding may also be involved in this bacterial-algal interaction (our interpretation). This is further confirmed by the identification of signaling molecules, including halogenated furanones, secreted by the algal culture with potential to inhibit QS and bacterial growth.

Microalgae can produce amino acids such as tryptophan, a precursor for auxin biosynthesis in bacteria, promoting mutualistic interactions [39,40].

The ubiquitous sulfur-containing dimethylsulfoniopropionate (DMSP) is a compound produced by phytoplankton [41], they are osmoregulatory and antioxidant substances in the algae that produce them [42]. DMSP leaked into the extracellular medium stimulates chemotaxis in some marine bacteria [43], which can metabolize it as a source of sulfur, carbon, and energy, in addition to stimulating the production of quorum-sensing molecules in bacteria [42].

On the other hand, QS signals may exert an algaecide effect in some microalgae. Dow (2021) has discussed the role of some signaling molecules such as alkyl-quinolones and tetramic acid (a spontaneous rearrangement of 3-oxo-AHLs) to inhibit photosynthesis (and microalgal growth), as well as being able to interrupt important cell cycle and metabolic activities in microalgae [33]. Segev et al. (2016) showed that during microalgae senescence, some species can release organic acids, such as p-coumaric acid, which can be recognized by bacteria and induce pathogenic responses [40].

Bacterial-produced QS inhibitors are suitable and efficient candidates to block the biological functions of QS microalgae, as they control algal blooms in water bodies so as to decrease concentrations of toxic species and prevent the harmful effects of toxins [44].

According to Zhang et al. (2020), bacterial QS signals are detected by microalgae that have developed several mechanisms to respond positively or negatively to these signals, which can be symbiotic or algaecide signals, depending on the species associations performed [44].

### 2.2. Oxygen and Carbon Dioxide Exchanges

The co-cultures of bacterium and algae may be effective in detoxifying inorganic and organic pollutants and removing nutrients from wastewater if compared to the activity of these microorganisms individually. Photosynthesis by cyanobacteria and eukaryotic algae provides oxygen, an essential factor for heterotrophic bacteria that degrade pollutants. Sequentially, the bacteria help the photoautotrophic growth of the collaborators, furnishing carbon dioxide and stimulating factors [17] and decreasing the oxygen concentration in the culture medium [15].

This relationship is interesting because microalgae can be part of a circular economy, since it allows the use of CO_2_ coming from industrial processes, like distilleries (CO_2_ from alcoholic fermentation and sugarcane bagasse burning), cement industry (CO_2_ from burning of energy source and CaCO_3_ decomposition), energy industry (CO_2_ from burning of energy source) as well as from aerobic and anaerobic treatment of wastewaters, as a carbon source to produce microalgae biomass. Simultaneously, the oxygen produced by phytoplankton may support the necessity of aerobic processes in these industries. The oxygen produced by microalgae could be used in an integrated process involving aerobic depuration of wastewaters, with a production of bacteria which could be used in different applications, like agricultural inoculants, among others, depending on the species of microorganism cultivated [45].

## 3. Effects of Interactions

### 3.1. Inhibitory Effect by Metabolites on Algae and Bacteria

Although several species of bacteria have a beneficial effect on algae growth, some bacterial species may also inhibit microalgae by producing extracellular algaecide compounds [22]. This inhibitory effect helps to control the proliferation of harmful algae in bodies of water [15].

Some bacteria can induce the lysis of microalgal cells. For instance, the *Kordia algicida* secretes an algaecide protease that hinders the growth of several diatomaceous marine species [46]. Bacteria can compete with microalgae for limiting nutrients, such as nitrogen and phosphorus, when they grow together in an organic carbon source. Due to the higher growth rate of bacteria than microalgae in this condition, their propagation would consume more nutrients, limiting the growth of eukaryotic photosynthetic microorganisms due to a lack of nutrients [25]. For instance, when *Acinetobacter* sp. was inoculated in the exponential phase, a substantial drop in the growth of *Botryococcus braunii* was observed. This bacterium, which presented a negative interaction with *B. braunii*, produces AHL signaling molecules involved in bacterial quorum sensing, which were found in the non-axenic culture of microalgae [47].

From these considerations, it is important to evaluate the environmental conditions of the processes where the presence of undesirable bacteria is minimized and preferentially suppressed to maximize the microalgal growth. Otherwise, favoring beneficial bacteria within an appropriate threshold is also a determinant for optimizing microalgae growth.

In contrast, *Tetraselmis* is a phytoplankton species that controls bacterial diseases in fish, inhibiting the growth of bacteria such as *Aeromonas hydrophila*, *Aeromonas salmonicida* strains NG and LL, *Lactobacillus* sp., *Serratia liquefaciens*, *Staphylococcus epidermidis*, *Vibrio anguillarum*, *Vibro salmonicida,* and *Yersinia ruckeri* type I [48]. This inhibitory effect against bacteria reported in this study can be a great ally for pharmacological treatments involving antibiotics, with the possibility of extracting bactericidal biomolecules produced by microalgae, and developing new drugs. Considering the huge environmental particularity conditions around the World, such findings suggest that there could be other microalgae producing substances that inhibit bacterial growth, highlighting the necessity of studies in this field.

### 3.2. Bacteria That Promote the Growth of Microalgae

Normally, non-pathogenic bacteria from several species have been found in microalgae cultivations with beneficial effects on their growth.

It has been shown that bacteria may modify the growth of phytoplankton, accumulating biomass and increasing cell productivity, which is of particular interest for industrial production. The bacteria of the genera *Alteromonas* and *Muricauda* allowed the most significant accumulation of *Dunaliella* biomass due to the increase in the availability of nitrogen for microalgae. However, more research is needed to understand the mechanisms behind these interactions [49].

In the cultivation of *Chlorella prototecoides* in synthetic wastewater media in co-culture with *Brevundimonas diminuta* with light intensities of 75 and 130 μmol photons m^−2^s^−1^, the μ values in non-axenic conditions were at least five times higher than in cultivations without co-culture. Thus, under these conditions, the addition of *Brevundimonas diminuta* was able to provide higher growth rates of *C. protothecoides* with more efficient nutrient removal [50].

A substantial promotion in the growth of phytoplankton has been described because the bacterium produces indole-3-acetic acid (IAA) [15]. The *Achromobacter* sp. produces IAA, which promotes the growth of *Haematococcus pluvialis*, with an increase in chlorophyll and cell concentrations [51]. In fact, the bacterial population in low concentrations can improve the microalgae metabolism by releasing factors that promote growth or reduce the concentration of O_2_ in the medium, preventing this gas from reaching an inhibitory concentration. As a consequence of higher microalgae growth in co-culture with bacteria, there is a higher removal of nutrients from the cultivation medium, which is particularly important in the tertiary treatment of wastewater since nitrogen and phosphorus are constituents of the microalgae biomass. In a study using synthetic wastewater with a semi-continuous process, the immobilized consortium of *Chlorella vulgaris* and *Azospirillum brasilense* led to an increase in the uptake of ammonium by culture [52]. Thiamine and tryptophan released by *C. sorokiniana* are signaling molecules that may be used by *A. brasilense* to synthesize and secrete another signaling molecule, indole-3-acetic acid, which promotes microalgae growth. The occurrence of signaling compounds, such as thiamine and tryptophan in the exudates of *Chlorella sorokiniana*, supports the mutualistic interaction of this photosynthetic microorganism with *A. brasilense* [53].

In a study where *C. vulgaris* was grown with *Bacillus pumilus*, it was found that the bacterium fixed nitrogen from the atmosphere producing ammonia, a nitrogen form that promoted the growth of the microalga [54]. It is also worth mentioning that there are nitrifying bacteria species able to convert ammonia in nitrate [55], which is another well-known source of nitrogen for microalgae, that requires the production of specific enzymes for nitrogen assimilation.

*Chlorella vulgaris* was also grown with four bacterial strains that may promote its growth: *Flavobacterium, Hypomonas, Rhizobium,* and *Sphingomonas* for 24 days, obtaining higher growth of the microalgae in co-culture with these bacteria (cell concentrations up to 3.31 g L^−1^) compared to the control (1.30 g L^−1^) [56]. The addition of bacteria in the cultivation of microalgae leads to a consortium in which the microorganisms exchange organic carbon, thus increasing the microalgae growth [12]. Rivas et al. (2010) observed that, at 20 °C, the presence of *Rhizobium* in the culture of *B. braunii*, led to an increase (approximately 50%) in the growth of this microalga. This positive interaction between *B. braunii* and *Rhizobium* sp. indicates the viability of using these bacteria as inoculum in large-scale cultures for producing microalgae biomass [47].

One of the most discussed topics in this area is the increase in algal biomass in non-axenic strains. Efforts are required to evaluate the behavior of each microalga species, mainly in relation to bacteria isolated in different places of the world, which give different environmental conditions, concerning parameters such as temperature, light cycles, and water source. Additional attempts regarding this topic include (i) the concentration of a bacterium species or mix of bacteria species present in the cultivation of microalgae in different proportions, (ii) supplementation of different compounds in the medium to support the bacterium growth aiming to favor the microalgae growth; and (iii) evaluation of bacterial extracts in culture medium for microalgae growth.

### 3.3. Supply of Nutrients

Microalgae may increase bacterial activity by secreting extracellular molecules such as lipids, proteins, and nucleic acids that serve as nutrients for bacterial growth. In this sense, dead microalgae cells can also provide nutrients for the growth of bacterial cells [15].

Croft et al. (2005) showed that vitamin B12 is an important molecule in algae metabolism, the main cofactor for methionine synthase, which depends on vitamin B12. They also observed that cobalamin auxotrophy had appeared numerous times throughout evolution processes, probably related to the presence or absence of vitamin B12-dependent enzymes. An example of this symbiosis is the case of bacteria of the genus *Halomonas*, that supply cobalamin for the microalgae *Amphidinium operculatum* [57]. *Pseudomonas* sp., on the other hand, produced a glycoprotein that performed as a growth factor for *Asterionella glacialis* [58]. In another research, the growth of *Chlorella* sp was shown to be improved due to the release of riboflavin by *E. coli* [59]. Moreover, bacterial communities may provide algae with inorganic micronutrients that could be invaluable in the environment [12].

One study showed that *Streptomyces* produced a respiratory inhibitor known as antimycin A (AA) that acts as mediator of Fe (III) in the halotolerant algae *D. salina*. AA forms fat-soluble complexes with Fe (III) that are absorbed and used by algae. Several species of phytoplankton can use bacterial siderophores as a source of Fe and at least one species of *Dunaliella, D. bardawil*, uses bacterial factors to improve iron acquisition. However, *D. salina* is unlikely to use AA as a mediator of Fe (III) in its natural habitat because *Streptomyces* species are not common in hypersaline environments characteristic of *D. salina*. The most likely interpretation of the results is that AA functions as an artificial mediator of Fe (III), forming Fe (III) -AA, which is a fat-soluble and stable complex that penetrates through the cell membrane and can be used directly by the cells, increasing Fe (III) bioavailability. Similar captures have been demonstrated for the use of organic lipophilic complexes of Cu and Ni ions by aquatic organisms [60].

The knowledge of nutrients demanded by microalgae and bacteria is essential to optimize their cell growth. The metabolomic approach may be a valuable tool for better understanding the interaction between these organisms, thus providing useful information to maximize the cell growth and production of metabolites by them. In fact, in co-cultures of microalgae and bacteria, the metabolomic study allows to detect the secretion of substances, such as vitamins, which are produced by one of these microorganisms and are essential for the other one.

### 3.4. Modification of the Composition of Microalgae in Co-Culture

Some studies have reported that the interaction with bacteria can also change the lipid content, especially the fatty acids of microalgae. Cho et al. (2015) reported an increase of 20% in triacylglycerols in *Chlorella vulgaris* cultivated in consortium with bacteria where it was detected the presence of *Hyphomonas* sp., *Flavobacterium* sp., *Sphingomonas* sp, and *Rhizobium* sp., with a significant increase in the proportion of palmitic (C16:0) and oleic (C18:1) acids if compared with results obtained with axenic cultivation [56].

The association of *Rhizobium sp*. KB10 with *B. braunii* increased algae growth by nine times and improved the oleate content, used to produce biodiesel [61]. Inoculation of the bacterial strain *Rhizobium* 10II in the cultivation of *Ankistrodesmus* sp. strain SP2-15 increased by 30% the chlorophyll content in the microalgae biomass, and the lipid productivity was up to 112 g.m^−2^d^−1^ on the sixth day of cultivation [62]. The co-cultivation of *Chlamydomonas reinhardtii* with *Bradyrhizobium japonicum* improved the growth of the microalgae by 3.9 times, reaching lipid contents 26% higher and increasing Fe-hydrogenase activity and H_2_ production [63].

In addition to the possibility of increasing the production of fatty acids for producing biodiesel, widely explored in the literature, it is interesting to conduct efforts in studies regarding the effect of several bacteria on the production of carotenoids by microalgae that produce these metabolites, which are used in food, cosmetics, and medicines. Although the classic forms to stimulate the synthesis of carotenoids are related to the exposition of the cells to a stress condition [64], there is not enough information about the influence of chemical compounds and bacteria on carotenogenesis. Metabolomic is an important tool that could help to understand the influence of bacteria on the lipid anabolism in microalgae, thus giving information about the growth conditions of xenic cultivations of microalgae to produce these compounds.

Studies demonstrated that the cultivation of *Scenedesmus obliquus* and the bacterial strains *Diaphorobacter* sp. and *Acidovorax facilis* positively influence the amount of microalgae biomass, and lipid production. Compared with the biomass of the axenic strain, the biomass of the co-culture of selected strains increased such cellular fraction by 3.5–24.8%. In addition, it was found that the bacteria adhered directly to the cell surface of *S. obliquus*, changing the concentration and composition of Extracellular Polymeric Substances (EPS) [65].

The consortium of *Chlamydomonas reinhardtii* CC-849 and *Azotobacter chroococcum*, under nitrogen deprivation, increased both the cell growth and the quantity of lipid in the microalga if compared with axenic microalga, which was attributed to the modulation of gene expression that regulates fatty acid metabolism. Such an increase in the lipid content of microalga was accomplished by a decrease in protein content [66]. Concerning this observation and considering the scarce information on the effect of co-culture in cell fractions other than lipids, it would be helpful to conduct studies aiming to analyze the centesimal composition of this biomass under such conditions, as well as to understand these effects inside the cell. It is worth to mention that for each cultivation medium for microalgae growth, there is a particular pH range and a particular salinity, thus, per se, selecting bacteria species which can grow in such conditions. Taking this consideration together with the microalga chosen to grow in such medium, the evaluation of its growth and composition in presence of each bacterium species can help in the selection of conditions favoring the growth of both species, including the adequate bacterial inoculum size that promote the microalgae growth and/or maximize the product of a valuable bioproduct. These studies are necessary because a major part of the microalgae biomass currently produced worldwide is under non-axenic conditions.

### 3.5. Flocculant Activity by Bacteria

The activity of bio-flocculant depends on the growth phase of the bacteria, being enzymatic activities related to the formation of bio-flocculants observed during the stationary growth phase. Although the production of bacterial bio-flocculant is beneficial for improving the formation of large flakes of microalgae and bacteria, the additional cost related to the carbon source required for the growth of these bacteria still remains a challenge [67], which evidence that organic by-products could be used to produce such bacteria, thus diminishing the cost of the process [68].

Exopolysaccharides and pyruvic and uronic acids are important for cell adhesion. In addition, factors such as the sources of nitrogen and carbon and the ratio between these two elements influence the production of bio-flocculants. The bio-flocculant produced by the *Paenibacillus polymyxa* exhibited high efficiency for the flocculation of *C. vulgaris* and *Scenedesmus* sp. [67].

Bacteria such as *Flavobacterium, Terrimonas,* and *Sphingobacterium* and their extracellular polymeric substances may help to increase the flocculating activity of algae such as *C. vulgaris*, resulting in sedimentable flakes [69].

Flocculation helps in the processes of microalgae biomass harvest, and the study of interactions between bacteria and microalgae which promote at the same time improvement of the microalgae growth and their flocculation, mainly at the end of the cultivation, could diminish the overall production cost. Therefore, depending on the use of the microalgae biomass, efforts need to be made to improve the production of non-toxic bio-flocculants at the end of cultivation, by the addition of bacteria that promote microalgae aggregation at the end of cultivation, or even by the addition of substances that increase the content of such components in the cells at this period.

### 3.6. Microalgae Co-Immobilization by Bacteria

Microalgae are part of the organisms attached to filters in wastewater treatment plants, where the wastewater percolates during the treatment process. In these filters, enzymes or whole cells may be immobilized, including microalgae cells, which serve to obtain more biomass and for removing macronutrients since the production of oxygen by the algae improves the aerobic degradation of these substances. Moreover, the consumption of CO_2_ and the production of exopolysaccharides by microalgae can increase the bacterial growth rate, as CO_2_ and the production of growth-promoting substances by bacteria can improve microalgae growth. However, bacteria and microalgae may produce substances that hinder the growth of the other co-immobilized organism. Besides, the increase in pH and oxygen concentration in the medium, due to photosynthetic activity, can reduce bacterial growth in the system with the co-immobilization of bacteria and algae [70].

When the nitrogen source was nitrate instead of ammonium, *C. vulgaris* co-immobilized with the bacterium *Azospirillum brasilense* strain Cd did not increase the growth compared with axenic *C. vulgaris* culture, but nitrogen uptake per cell improved. The initial ammonium concentration affects its uptake by immobilized cells of *Chlorella vulgaris* alone, but not when this photosynthetic microorganism is co-immobilized with *A. brasilense*. Nitrogen uptake depends more on the number of *Chlorella vulgaris* cells than on the initial concentration of nitrogen [71]. On the other hand, under heterotrophic conditions, the co-immobilization of *Chlorella* spp. with *Azospirillum brasilense* promoted the starch consumption in *Chlorella* spp. This phenomenon may occur because the bacteria produce indole-3-acetic acid (IAA), inducing cell growth of *Chlorella* spp. [52,60]. It is worth mentioning that investigations about other nutrients, temperature, pH value, and other factors generally related to cell immobilizations can be applied to co-immobilization, and it would be interesting to perform studies considering diffusional effects, cell death, cell release, that can provide information about the system, thus providing security for implantation of industrial plants.

## 4. Microalgae as Potential Raw Material for Bioproducts

Considering the information on the interaction of microalgae and bacteria, besides the high potential of using microalgal biomass as a source of carbohydrates or fatty acids for energy production, food, cosmetic, and pharmaceutical industries, one could develop products in which microalgae, or their components could be used to confer special properties to them (Figure 2).

### 4.1. Extracts for Microbial Growth

Considering that bacteria and microalgae are rich in valuable organic compounds, and considering the related well-succeeded co-culture between these organisms, their extract could improve their growth. In this approach, Carvalho et al. (2021) developed a bacterial extract to promote the cultivation of microalgae, providing important nutrients for their growth. This extract allows greater growth in axenic and non-axenic strains of *Dunaliella salina* and non-axenic strains of *Chlorella vulgaris* [68]. Otherwise, a microalgae extract was developed from the ratio of consumption of organic matter from algae by bacteria. Since there are different species of microalgae with different protein compositions, it is handy for replacing animal protein in a culture medium used in the production of vaccines and biopharmaceuticals because it is possible to develop a specific culture medium culture for several bacteria species [72].

### 4.2. Bioproducts

#### 4.2.1. Microalgae for Developing Prebiotic Products

It is possible to develop foods based on the interactions between microalgae and bacteria, such as those that are not digestible, to beneficially affect the host by stimulating proliferation or activity of populations of beneficial bacteria in the intestine. There was a positive effect of *Arthrospira* on bacteria in the intestine, increasing the growth rate of *Lactobacillus*, thus supporting the function of the digestive tract and being used as a prebiotic [73]. For example, good results have been observed employing oligosaccharides, resulting from the hydrolysis of polysaccharides from *Chlorella vulgaris* and *Arthrospira platensis* as they increased the in vitro growth of beneficial bacteria such as *Bifidobacterium* animalis and *Lactobacillus casei*, thus acting as prebiotics [74]. In addition, these authors relate that polyssacharides from algae can be a source of galactooligosaccharides (GOS) and xylooligosaccharides (XOS). The laminarins in algae genera such as *Saccharina* and *Laminaria*, were recognized as dietary fibers with prebiotic properties [74,75]. Considering the potential of microalgae to promote growth of beneficial intestinal bacteria, it would be helpful to carry out studies aiming to determine if these phenomena can be evidenced in vivo, with an evaluation of the benefits in human diets using permitted daily quantity of microalgae considered GRAS (Generally Recognized As Safe) by the United States Food and Drug Administration (US-FDA) [76] or safety by others Regulatory Authorities, for example European Food Safety Authority (UE—EFSA) [77].

High-fat diet of rats supplemented with 95% ethanol extract of *Arthrospira platensis* (APL95) led to a hypolipidemic effect if compared with high-fat diet without supplementation of the extract (HFD). In fact, the total cholesterol and triglycerides were 120.2% and 27.8% higher in HFD than in APL95 diet. APL95 diet also had a protective effect on liver in rats treated with high-fat diet. Concerning gut microbiota, APL95 diet promoted an increase in beneficial bacteria, such as *Alloprevotella, Araprevotella, Barnesiella, Porphyromonadaceae* and *Prevotella* as well as a decrease in undesired microorganisms, including *Clostridium* XVIII, *Olsenella, Phascolarctobacterium, Romboutsi* and *Turicibacter*. Therefore, APL95 diet could be used as a new adjunctive therapy and functional food to regulate the intestinal microbiota in diabetic and obese people [78].

#### 4.2.2. Animal Diet

Cerezuela et al. (2012) carried out in vivo studies of experimental diets with *Tetraselmis chuii* (T), *Phaeodactylum tricornutum* (P) and *Bacillus subtilis* (B), simple or combined, showing morphological changes and significant signs of intestinal damage. The diets applied to fish led to a decrease in the bacterial diversity in the intestinal microbiota. Only diets containing *Bacillus subtilis* resulted in a significant reduction in the height of the microvilli. Moreover, fish fed with experimental diets showed different signs of edema and inflammation, and the authors concluded that such effects could compromise fish body homeostasis [79]. These findings highlight the necessity of evaluating, case by case, the benefits and risks of including any microorganism in animal and human diets.

Concerning strict biochemical findings, studies show that chickens fed with microalgae *Porphyridium* sp. in a proportion of 5% and 10% of its standard diet led to a reduction of 11% and 28% in the level of serum cholesterol, respectively, accomplished by a reduction of 10% in consumption of food, without any loss in production of eggs if compared to the control. Moreover, when microalgae biomass was included in the diet, egg yolks presented a tendency to have lower levels of cholesterol (by 10%), higher levels of linolenic (by 29%) and arachidonic acids (by 24%). Their color became darker as a result of the higher levels of carotenoids (2.4 times more with chicken fed with 5% supplement), thus encouraging the development of feed for chickens with polyunsaturated fatty acids and dietary fibers [80]. According to Becker (2004), the incorporation of algae in poultry feed offers a promising perspective and it is estimated that about 30% of the current world production of algae is sold for application in animal feed [6]. Considering such findings, it is worth developing studies concerning the incorporation of microalgae in feed destined to poultry, cattle, swine, fish production, which could allow an increment of valuable biological compounds in products from these animals [81].

It is worth mentioning that metabolomic studies could evaluate the compounds produced by intestinal bacteria in a diet containing microalgae and/or their extracts, being an important field to be considered in further research. For these studies, it could be used in vitro models or even in vivo evaluation, employing microalgae biomass (in safe quantities allowed by Regulatory Authorities). Such studies can provide information for the development of healthier processed foods, in which the incorporation of microalgae biomass could improve the consumer’s health, not only furnishing beneficial nutrients but also increasing the beneficial microorganisms in the gut.

### 4.3. Cosmetics

The growing need for obtaining safe products by bioprocesses has made microalgae a sustainable source for new products. Currently, microbial sources are the best available on the market to replace implemented entities [82].

Several secondary metabolites produced by algae are known to benefit the skin. Algae cells are naturally exposed to oxidative stress, which makes them develop efficient protection systems against radicals and reactive oxygen species, producing biomolecules that may act in cosmetics against the damaging effects of UV radiation, promoting the same action of inorganic filters and organic agents currently commercialized. There is an increase in the production of carotenoids and chlorophyll by *Chlorella vulgaris, Arthrospira,* and *Nostoc* when growing in the presence of radiation. These biomolecules can help to protect against the oxidative process of oil in formulations, especially in emulsions with a large quantity of oily phase, as they have antioxidant activities [83]. Such properties of microalgae can be used for the development of sunscreens, being associated with the formulation or the skin.

Due to the fact that biofilm is related to infections, particularly due to the low susceptibility of microorganisms to traditional antimicrobial agents, microalgae may be explored seeking an innovation to solve this problem. The antibiofilm activity of *Arthrospira platensis* extracts, which are abundant in free fatty acids, was verified. The nanocarriers based on copper alginate loaded with extract, were able to inhibit the formation of biofilms from one and two species of *Cutibacterium acnes*, but did not inhibit preformed biofilms. Nanovectorized extracts reduced the growth of *Candida albicans* biofilms, as well as preformed biofilms [84].

Proteins are molecules currently present in large amounts in microalgae and they can be used as alternatives to proteins from plants and animals in cosmetic formulations, which opens a large research field with high industrial potential. Polysaccharides and carotenoids are other macromolecules present in microalgae with high potential to be used in cosmetic formulations. Such compounds can act on the cosmetic formulation, changing their properties, protecting and or stabilizing them. These molecules can also act in skin, avoiding its water loss and/or nourishing it.

Considering that cosmetic formulations are destined to several uses, the change of the source of raw material can, in addition to the change in formulations properties, affect the conservation of the formulations and the microbiota of the skin. Research demonstrates that the levels of sebum and hydration are essential factors for the diversity of the skin’s microbiota, differing according to age and skin groups with acne [85]. This field is to be evaluated with the use of formulations incorporating microalgae and their extracts in cosmetic formulations.

### 4.4. Pharmaceuticals

Currently, resistant strains are gaining attention in the treatment of bacterial infections [10]. Therefore, a new strategy is oriented, using a chemically modified *Chlamydomonas reinhardtii* as a drug delivery system. These modified microalgae masked vancomycin thanks to an insertion of a photocleavable binder on the cell surface, and the antibiotic was released in a controlled way, under exposure to ultraviolet light (340–400 nm). This technique was tested on *Bacillus subtilis*, successfully resulting in growth inhibition of this bacterium [86]. Microalgae, as earlier commented, can produce compounds that lead to the inhibition of bacteria, which can be extended to other microorganisms and even viruses [87]. Thus, these photosynthetic microorganisms present great potential to source new antibiotics for therapeutic uses. Projects involving the screening of antibiotic-producing microalgae deserve worldwide efforts, taking advantage of a significant number of currently available microalgae collections.

#### 4.4.1. Antibacterial Extracts—Compounds Produced by Microalgae

Although the presence of pathogenic bacteria has not been reported in microalgae cultures, the production of antibiotics by microalgae could contribute to the biological safety of the obtained biomass.

The antioxidant activity of *Dunaliella salina* extract can inhibit the growth of opportunistic bacteria such as *E. coli* and *Pseudomonas aeruginosa*. In liquid medium, the growth of these two bacteria was delayed by 263% and 20%, respectively. Conversely, this same extract did not affect the growth of *Klebsiella pneumoniae*, *Enterococcus* sp., and *Staphylococcus aureus*, demonstrating the specificity in the action of the whole cell extract [88].

It was found that the extracts of fatty acid mixture from *Chlorella vulgaris* showed antibacterial activity against *Salmonella Typhi*, a Gram-negative bacterium, the causative agent of enteric fever, infectious diarrhea, and sepsis in humans, in addition to *E. coli* and *P. aeruginosa* ATCC 29212. The antimicrobial activity of *Chlorella vulgaris* was found to control Gram-positive bacteria such as *Bacillus subtilis, Enterococcus faecalis* ATCC 29212, *Streptococcus pyogenes* ATCC 12344, and *Staphylococcus aureus* ATCC 25923 [89].

#### 4.4.2. Removal of Dental Biofilm by Algal Extract

There was a report in the literature of the alga *Chlamydomonas reinhardtii* inhibiting bacterial aggregation of *Escherichia coli*, leaving the bacteria vulnerable to predation [90]. In addition, there is an interesting disaggregating effect to be used in biofilms harmful to human health.

Extracts were prepared from 225 microalgae, containing polyunsaturated fatty acids with antibacterial and antibiofilm properties [91]. Vishwakarma and Vavilala (2019) evaluated the sulfated polysaccharides extracted from *Chlamydomonas reinhardtii* against *Bacillus subtilis, Streptococcus* sp., *Neisseria mucosa* and *E. coli*, validated the antibacterial activity in vitro, and showed that they can prevent biofilm formation and eradicate pre-formed biofilms [92].

## 5. Conclusions

Processes based on interaction of microalgae and bacteria are recognized as a renewable and sustainable technology, in compliance with current economic and low-carbon guidelines for environmental protection and resource recovery. Consortium-based technologies are expected to be viable for the current microalgae industry. Nevertheless, more research is required with an in-depth evaluation of the mechanisms of interaction between these microorganisms, as well as large-scale cultures for biotechnological application in actual conditions [40].

In nature, regardless of the environmental condition, microalgae and bacteria interact through communication by quorum sensing, which may change the metabolism of both organisms. The understanding of the interaction between different species of bacteria and microalgae may highlight the production of new molecules by microalgae and even improve the production of already known molecules. Moreover, the knowledge of the structure of these molecules and their actions on other organisms, including bacteria, indicates that the use of extracts or purified molecules from microalgae can be useful in the development of health, food, and cosmetic products, as well as in the development of nutritional ingredients for culture media. Efforts are necessary to develop such useful products, increasing the world microalgae demand, and contributing to the sustainable production of industrial products.

As future perspectives, we can indicate (i) isolation of new antibiotic biomolecules from microalgae and carry out computational studies; (ii) development of pharmacotechnical formulations for pre-clinical studies; (iii) development of dermatological formulations for the treatment of acne; (iv) development of skincare creams and/or gels; (v) development of microalgae-based prebiotic juices for bowel regulation; (vi) development of specific culture media for the growth of heterotrophic microorganisms, mainly focusing on vaccines and biopharmaceuticals.

## Figures and Tables

**Figure 1 microorganisms-10-02029-f001:**
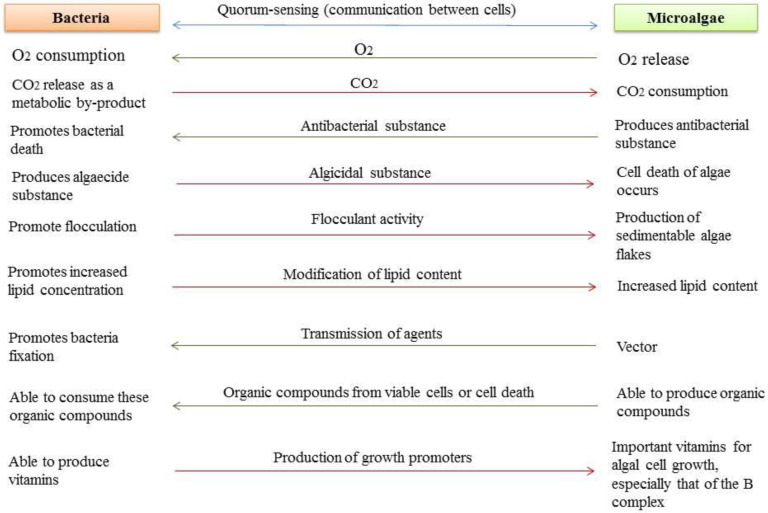
Relationship between bacteria and microalgae indicating its effects.

**Figure 2 microorganisms-10-02029-f002:**
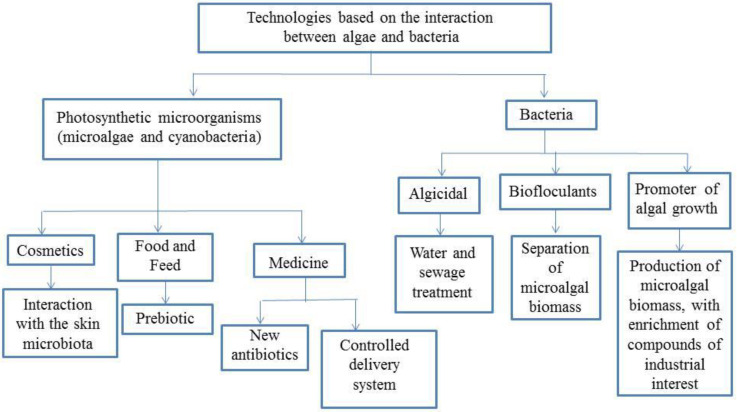
Diagram about how the study of microalgae-bacteria interactions can result in bioproducts.

## Data Availability

Not applicable.

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
