# Peer review of "Obtaining Bioproducts from the Studies of Signals and Interactions between Microalgae and Bacteria"

_microorganisms, 2022, doi:10.3390/microorganisms10102029_

Round 1
Reviewer 1 Report
Reviewer comments
Manuscript: Obtaining bioproducts from the studies of signals and interac-
tions between microalgae and bacteria
ID: microorganisms-19482415
Date: 2022-09-21
General comments
The author proposed a review of the recent investigations tackling the question of the interaction (cooperation/competition) between microalgae and bacteria. Their focus is biotechnology-oriented (not ecophysiology) with a clear aim at easing high-added value molecule production. The manuscript structure is sound, and the overall coverage of the topic is adequate for publication. Still, some aspects have to be improved, especially to reinforce the readability of this work (see Major concerns). Therefore, this reviewer advises for publication once the following concerns are addressed.
Major concerns
1. The introduction suggests the use of photoautotrophy to produce a large number of molecules of interest for the pharmaceutical/nutraceutical industry. In my opinion, this way of presenting the stake is misleading. These two industries resort to heterotrophy to produce the biomolecules they seek. Furthermore, the overall yield of large-scale outdoor photoautotrophic cultures is low. This argument is a double sword: it reinforces the need to better understand bacteria/microalgae interactions (to improve the yields), but also shows that profitability may simply be out of reach. I strongly advise the authors to comment on these two points in the introduction.
2. The wording used is sometimes unnecessarily lengthy or simply descriptive (with very limited added value), and some paragraphs do not deal with bacteria/microalgae interaction or pharmaceutical/nutraceutical molecules. They should be improved or simply removed.
List: l173 – l179 (biofuel), section 2.2 $2 (no added value, too general), l305 – l335 (too descriptive, add a comment on bacterial nitrification or diazotrophy), section 3.5 $1 (add numbers to support the claim), section 4.2.1 (remove, no added value), section 4.2.3 (remove, no link with bacteria / microalgae interaction), section 4.2.5 (remove, unrelated to the topic), section 4.3.1 (unrelated, remove)
Minor concerns
Please be more specific l509, you refer to the US-FDA.
Please remove all mention of Spirulina, this is the commercial name of Arthrospira.
Numerous strain names are not in italic, please correct. List lines: 40, 181, 214, 264, 284, 285, 305, 380, 426, 490, 496, 500, 509, 514, 560, 561, 584.
Figure line 2, microalgae side, there is a typo.
Author Response
Dear Reviewer,
Thank you for the opportunity to revise our manuscript microorganisms - 1948241 entitled "Obtaining bioproducts from the studies of signals and interactions between microalgae and bacteria". We appreciate the referees´ comments and, considering them, all the alterations were marked up using the “track changes” function in the manuscript. Please, find below the answers to all comments.
General comments
The author proposed a review of the recent investigations tackling the question of the interaction (cooperation/competition) between microalgae and bacteria. Their focus is biotechnology-oriented (not ecophysiology) with a clear aim at easing high-added value molecule production. The manuscript structure is sound, and the overall coverage of the topic is adequate for publication. Still, some aspects have to be improved, especially to reinforce the readability of this work (see Major concerns). Therefore, this reviewer advises for publication once the following concerns are addressed.
- Major concerns
The introduction suggests the use of photoautotrophy to produce a large number of molecules of interest for the pharmaceutical/nutraceutical industry. In my opinion, this way of presenting the stake is misleading. These two industries resort to heterotrophy to produce the biomolecules they seek. Furthermore, the overall yield of large-scale outdoor photoautotrophic cultures is low. This argument is a double sword: it reinforces the need to better understand bacteria/microalgae interactions (to improve the yields), but also shows that profitability may simply be out of reach. I strongly advise the authors to comment on these two points in the introduction.
ANSWER: Thank you for the notes. In the last paragraph of the introduction, we rearranged and recognized it. We included the following text “...the success in cultivating microalgae under heterotrophic conditions is recognized, but in this context, this type of cultivation has the disadvantage of the need for axenic culture, and any contamination by heterotrophic microorganisms is strictly prohibited because they grow faster than microalgae. From the productive point of view, we aim to increase the cell concentration and these biomolecules, within stable, safe and robust processes, in order to guarantee the quality…”, demonstrating that we do not ignore this possibility, but that there is a way to improve photoautotrophic crops using the concept of co-culture or technologies made from this knowledge (lines 74-88 of the revised manuscript).
- The wording used is sometimes unnecessarily lengthy or simply descriptive (with very limited added value), and some paragraphs do not deal with bacteria/microalgae interaction or pharmaceutical/nutraceutical molecules. They should be improved or simply removed. List: l173 – l179 (biofuel), section 2.2 $2 (no added value, too general), l305 – l335 (too descriptive, add a comment on bacterial nitrification or diazotrophy), section 3.5 $1 (add numbers to support the claim), section 4.2.1 (remove, no added value), section 4.2.3 (remove, no link with bacteria / microalgae interaction), section 4.2.5 (remove, unrelated to the topic), section 4.3.1 (unrelated, remove)
ANSWER: Thank you for all these observations. Section 2.2 was renamed to “Oxygen and Carbon dioxide exchange”, and for a better understanding by the readers, it was rewritten and shortened (line 227 of the revised manuscript). The last paragraph of section 3.3 (new section 3.2) “Bacteria that promote the growth of microalgae” were also revised and shortened (lines 279-335 of the revised manuscript). Sections 4.2.1 (Food supplements and animal feed), 4.2.2 (“Prebiotics benefits human diet”), and 4.2.3 (“Prebiotics in disease prevention”) were revised and merged in a new section 4.2.1 (Microalgae for developing prebiotic products) (lines 495-523 of the revised manuscript). Section 4.2.5. was completed removed as suggested. The figure 2 was corrected accordingly. Furthermore, we added a comment about bacterial nitrification (lines 313-316 of the revised manuscript).
- Minor concerns
Please be more specific l509, you refer to the US-FDA.
ANSWER: Thank you for this observation. We were more specific, as indicated: “GRAS (Generally Recognized As Safe) by the United States Food and Drug Administration (US-FDA) [76] or safety by others Regulatory Authorities, for example, European Food Safety Authority (UE - EFSA)” (lines 511 - 513 of the revised manuscript).
- Please remove all mention of Spirulina, this is the commercial name of Arthrospira.
ANSWER: All mentions of “Spirulina” were removed and replaced by “Arthrospira” as directed. The changes were made throughout the text using the “track changes” function.
- Numerous strain names are not in italic, please correct. List lines: 40, 181, 214, 264, 284, 285, 305, 380, 426, 490, 496, 500, 509, 514, 560, 561, 584.
ANSWER: Thank you for detecting the mistakes. All the scientific names are now written in italic form. We used the “track changes” function to alter it along the text.
- Figure line 2, microalgae side, there is a typo.
ANSWER: Thank you for detecting this mistake. In the revised manuscript, “Consumo CO2” was replaced by “CO2 consumption” (Figure 1, line 132 of the revised manuscript).
Reviewer 2 Report
Review. Obtaining bioproducts from the studies of signals and interactions between microalgae and bacteria
Évellin do Espirito Santo, Marina Ishii, Uelinton Manoel Pinto, Marcelo Chuei Matsudo and João Carlos Monteiro de Carvalho
The first part of the manuscript review in detail reports on the interaction between some microalgae strains and bacteria. Benefits of this type of interactions that are typical in non-axenic (or outdoor) culture of microalgae were described. The novelty of this review should be described because there are similar reports in the last five years.
In the second part, products from microalgae culture are described. The relationship between the first and the second part is not clear, and this section is not related with what be expected from the title and the abstract section.
The abstract mentioned that the manuscript addresses the development of new pharmaceutical, cosmetic and food bioproducts from microalgae based on these evaluations. However, the products are those typically obtained from microalgae.
The manuscript could be considered for publication after some changes:
An abstract related with the contents.
The name of the strains must be in italics (lines 181, 305, 380, 426, 569)
It would be expected that the manuscript describes how to implement co-culture in practice, for the production of valuable bioproducts.
The metabolomic approach as a useful tool for understanding of the interaction between these organisms was mentioned more than once in the manuscript. Please explain how this could be done.
Line 452, please explain the meaning of “the nitrogen source was altered from ammonium to nitrate”.
Line 461, please explain how bacteria alter the metabolic pathways in the microalga.
Line 486, why different protein compositions are useful for replacing animal protein in the production of vaccines and biopharmaceuticals?
Author Response
Dear Reviewer,
Thank you for the opportunity to revise our manuscript microorganisms - 1948241 entitled "Obtaining bioproducts from the studies of signals and interactions between microalgae and bacteria". We appreciate the referees´ comments and, considering them, all the alterations were marked up using the “track changes” function in the manuscript. Please, find below the answers to all comments.
- An abstract related to the contents.
ANSWER: Thank you for this note. We restructured the text to be more specific (lines 12- 25 of the revised manuscript).
- The name of the strains must be in italics (lines 181, 305, 380, 426, 569)
ANSWER: Thank you for detecting the mistakes. All the scientific names are now in written italic form. The changes were made throughout the text using the “track changes” function.
- It would be expected that the manuscript describes how to implement co-culture in practice, for the production of valuable bioproducts.
ANSWER: The authors acknowledge the reviewer for such important consideration. We described in the revised manuscript an example of how we can implement co-culture in practice at the end of the item 3.5. “It is worth to mention that… a valuable bioproduct” (lines 412-419 of the revised manuscript).
- The metabolomic approach as a useful tool for understanding of the interaction between these organisms was mentioned more than once in the manuscript. Please explain how this could be done.
ANSWER: Thank you for your observation, which allowed us to improve the manuscript. In the revised manuscript, we included examples of how the metabolomics study could be used, either to optimize cell growth (lines 364-370 of the revised manuscript) or to develop new healthy products (lines 549-556 of the revised manuscript).
- Line 452, please explain the meaning of “the nitrogen source was altered from ammonium to nitrate”.
ANSWER: The authors acknowledge the reviewer for this query. The phrase was rewritten, aiming to clarify the information. “When the nitrogen source was nitrate instead of ammonium...” (line 458 of the revised manuscript).
- Line 461, please explain how bacteria alter the metabolic pathways in the microalga.
ANSWER: Thank you for showing a way to improve this phrase about this topic. It was rewritten as: “On the other hand, under heterotrophic conditions, the co-immobilization of Chlorella spp. with Azospirillum brasilense…” (lines 464-467 of the revised manuscript)
- Line 486, why different protein compositions are useful for replacing animal protein in the production of vaccines and biopharmaceuticals?
ANSWER: Thank you for your inquiry. Microalgal protein extract may be used in culture medium for microbial growth. Since different microorganisms have different nutritional needs, it is possible to develop specific culture medium. This information was added in the revised manuscript (lines 490-493 of the revised manuscript)
At your disposal for any additional necessary information,
Looking forward to hearing from you,
Yours faithfully,
João Carlos Monteiro de Carvalho
Reviewer 3 Report
Here are the comments
1. The abstract is so general that anyone working in microalgae biomass areas can figure out these points without reading your draft.
2. The introduction section is poorly written. The starting paragraphs failed to guide the readers to the question you wanted to raise. For example, In lines 46 to 48, the authors said, "However, the productivity of carotenoids from microalgae has historically been very low, and the economic feasibility of algae biotechnology is limited by photosynthetic efficiency and processing costs." Then you just jumped to "the production of microalgae and cyanobacteria can be negatively impacted by bacteria, fungi, and viruses (in line 49)". Is there any link between the low productivity of carotenoids from microalgae and the negative impacts of bacteria, fungi, or viruses? Are they the major impacts? The authors failed to provide a logical transition between these sentences and items.
3. The authors addressed, "The production of microalgae and cyanobacteria can be negatively impacted by bacteria, fungi, and viruses (in line 49)". Then why did you want to investigate the interaction between microalgae and bacteria, other than microalgae with fungi or viruses? The authors did not give a reasonable explanation.
4. Section 2 is entitled "Microalgae-bacteria interaction." However, the authors failed to provide a clear definition of Microalgae-bacteria interaction. It seems that section 2.1 is about to tell readers how to detect or quantify the interactions, but I do not know why the authors provided section 2.2. Can you use the oxygen and CO2 ratio to represent the interactions between microalgae and bacteria? The author did not give a clear presentation here.
5. Section 3 is organized in chaos. Section 3.1. and 3.2. can be combined. Section 3.2 can even be neglected since the major topic of the draft is still microalgae. Section 3.3, as the authors addressed in their introduction, "Bacteria that promote the growth of microalgae," has been well studied and widely reviewed. I saw no novelty here. Section 3.4-3.7 also did not provide how the interactions "focuses on the production process of cell culture applied to the production of high value-added biomolecules in biopharmaceuticals (line 69-71)". Besides, the authors just listed the results but failed to show how they " carefully analyzed interaction mechanisms in co-cultures (lines 71 and 72)".
6. Section 4 is hard to understand. Why don't you just focus on how the interactions between microalgae and bacteria improve the production of value-added bioproducts from microalgae? Like by increasing biomass production? By changing bioproduct production selectivity? Or by producing new bioproducts? This section should be a major part of your draft, other than simply listing the information like what you did in section 4.2.1 or 4.3.2.
7. English is poor.
8. Relatively old references, only roughly 1/3 of the 93 references were published in the past five years.
Author Response
Dear Reviewer,
Thank you for the opportunity to revise our manuscript microorganisms - 1948241 entitled "Obtaining bioproducts from the studies of signals and interactions between microalgae and bacteria". We appreciate the referees´ comments and, considering them, all the alterations were marked up using the “track changes” function in the manuscript. Please, find below the answers to all comments.
- The Abstract is so general that anyone working in microalgae biomass areas can figure out these points without reading your draft.
ANSWER: Thank you for this observation. The Abstract was restructured to be more specific (lines 12-25 of the revised manuscript).
- The introduction section is poorly written. The starting paragraphs failed to guide the readers to the question you wanted to raise. For example, In lines 46 to 48, the authors said, "However, the productivity of carotenoids from microalgae has historically been very low, and the economic feasibility of algae biotechnology is limited by photosynthetic efficiency and processing costs." Then you just jumped to "the production of microalgae and cyanobacteria can be negatively impacted by bacteria, fungi, and viruses (in line 49)". Is there any link between the low productivity of carotenoids from microalgae and the negative impacts of bacteria, fungi, or viruses? Are they the major impacts? The authors failed to provide a logical transition between these sentences and items.
ANSWER: Thank you for this observation. We put together the second and third paragraphs, in which we try to connect the fact that efforts are still lacking to overcome bottlenecks in the process, so that the microalgae are protagonists, and how other microorganisms in symbiosis can help to improve this technology (lines 44-53 of the revised manuscript).
- The authors addressed, "The production of microalgae and cyanobacteria can be negatively impacted by bacteria, fungi, and viruses (in line 49)". Then why did you want to investigate the interaction between microalgae and bacteria, other than microalgae with fungi or viruses? The authors did not give a reasonable explanation.
ANSWER: Thank you for the comment. The corresponding paragraph was complemented with the sentence “...On the other hand, other species may have a positive impact, as it takes place in nature.”. By the present review, we propose that it is important to comprehend both situations (negative and positive impact) of other microorganisms’ species for enabling optimum conditions for microalgae biomass production, favoring the beneficial species (lines 50-58 of the revised manuscript).
- Section 2 is entitled "Microalgae-bacteria interaction." However, the authors failed to provide a clear definition of Microalgae-bacteria interaction. It seems that section 2.1 is about to tell readers how to detect or quantify the interactions, but I do not know why the authors provided section 2.2. Can you use the oxygen and CO2 ratio to represent the interactions between microalgae and bacteria? The author did not give a clear presentation here.
ANSWER: Thank you for your notes. Concerning section 2.2, we intended to show the exchange between microalgae and bacteria so that while one produces CO2, the other consumes it, and an inverse relationship occurs with O2. In this case, we changed the title to “Oxygen and carbon dioxide exchanges” (line 227 of the revised manuscript).
- Section 3 is organized in chaos. Section 3.1. and 3.2. can be combined. Section 3.2 can even be neglected since the major topic of the draft is still microalgae. Section 3.3, as the authors addressed in their introduction, "Bacteria that promote the growth of microalgae," has been well studied and widely reviewed. I saw no novelty here. Section 3.4-3.7 also did not provide how the interactions "focuses on the production process of cell culture applied to the production of high value-added biomolecules in biopharmaceuticals (line 69-71)". Besides, the authors just listed the results but failed to show how they " carefully analyzed interaction mechanisms in co-cultures (lines 71 and 72)".
ANSWER: Thank you for the consideration. Sections 3.1. and 3.2. were combined into a single topic (new section: 3.1 “Inhibitory effect by metabolites on algae and bacteria”, lines 248-278 of the revised manuscript). The last four paragraphs in section 3.3 (new 87section 3.2, lines 279-335 of the revised manuscript) have been restructured into two paragraphs and provide a framework for discussing sections 4.1, 4.2., and 4.3. Sections 3.4-3.7 focus on the different interactions between bacteria and microalgae in the cell culture production process applied to the production of biomolecules to support the theory that new products can be based on it (new sections 3.3 - 3.6, lines 336, 371, 421, 445 of the revised manuscript). Referring to lines 69-71, we have rewritten the phrase to clarify the object of this review (lines 77-88 of the revised manuscript).
- Section 4 is hard to understand. Why don't you just focus on how the interactions between microalgae and bacteria improve the production of value-added bioproducts from microalgae? Like by increasing biomass production? By changing bioproduct production selectivity? Or by producing new bioproducts? This section should be a major part of your draft, other than simply listing the information like what you did in section 4.2.1 or 4.3.2.
ANSWER: Thank you for all your notes. Sections 4.2.1 (Food supplements and animal feed), 4.2.2 (“Prebiotics benefits human diet”), and 4.2.3 (“Prebiotics in disease prevention” were revised and improved, merging in one new section 4.2.1 (Microalgae for developing prebiotic products, line 482 of the revised manuscript).
- English is poor.
ANSWER: Thank you for the comment. The manuscript was revised by a fluent English speaker, and the corrections were marked up with “track changes” function.
- Relatively old references, only roughly 1/3 of the 93 references were published in the past five years.
ANSWER: The old references are the basis for this area of co-cultures. However, to bring more recent references, we removed these references:
Schwenk, D.; Nohynek, L.; Rischer, H. Algae-Bacteria Association Inferred by 16S RDNA Similarity in Established Microalgae Cultures. Microbiologyopen 2014, doi:10.1002/mbo3.175.
Barka, A.; Blecker, C. Microalgae as a Potential Source of Single-Cell Proteins. A Review. Biotechnol. Agron. Soc. Environ. 2016.
Navacchi, M.F.P.; de Carvalho, J.C.M.; Takeuchi, K.P.; Danesi, E.D.G. Desenvolvimento de Bolo de Mandioca Enriquecido Com Spirulina platensis e Farelo de Fecularias. Acta Sci. - Technol. 2012, 34, 465–472, doi:10.4025/actascitechnol.v34i4.10687.
Kumoro, A.C.; Johnny, D.; Alfilovita, D. Incorporation of Microalgae and Seaweed in Instant Fried Wheat Noodles Manufacturing: Nutrition and Culinary Properties Study. Int. Food Res. J. 2016, 23, 715–722.
Cardoso, T.; Esmerino, L.A.; Bolanho, B.C.; Demiate, I.M.; Danesi, E.D.G. Technological Viability of Biobased Films Formulated with Cassava By-Product and Spirulina platensis. J. Food Process Eng. 2019, 42, 1–11, doi:10.1111/jfpe.13136.
Wichuk, K.; Brynjólfsson, S.; Fu, W. Biotechnological Production of Value-Added Carotenoids from Microalgae: Emerging Technology and Prospects. Bioengineered 2014, 5, 204–208, doi:10.4161/bioe.28720.
Magdouli, S.; Brar, S.K.; Blais, J.F. Co-Culture for Lipid Production: Advances and Challenges. Biomass and Bioenergy 2016.
Furthermore, we added these more recent references:
Morales-Sánchez, D.; Martinez-Rodriguez, A.O.; Matinez, A. Heterotrophic cultivation of microalgae: production of metabolites of commercial interest. J. Chem. Technol. Biotechnol. 2017, 92, doi: 10.1002/jctb.5115.
Kholany, M.; Coutinho, J.A.P.; Ventura, S.P.M. Carotenoid Production from Microalgae: The Portuguese Scenario. Molecules 2022, 27, 2540. doi: 10.3390/molecules27082540.
Xi, H.; Zhou, X.; Arslan, M.; Luo, Z.; Wei, J., Wu, Z.; Gamal El-Din, M. Heterotrophic nitrification and aerobic denitrification process: Promising but a long way to go in the wastewater treatment. Sci. Total Environ. 2022, 805, 150212. doi:10.1016/j.scitotenv.2021.150212
Bature, L.; Melville, K.M.; Rahman, P. Microalgae as feed ingredients and a potential source of competitive advantage in livestock production: A review. Livest. Sci. 2022, 259, 10490, doi: 10.1016/j.livsci.2022.104907.
de Oliveira, A. P.; Bragotto, A. Microalgae-based products: Food and public health. 2022, 6. 100157. doi: 10.1016/j.fufo.2022.100157.
Markou, G.; Imene, C.; Tzovenis, J. Microalgae and cyanobacteria as food: Legislative and safety aspects. Cultured Microalgae for the Food Industry, 2021. doi:10.1016/B978-0-12-821080-2.00003-4
At your disposal for any additional necessary information,
Looking forward to hearing from you,
Yours faithfully,
João Carlos Monteiro de Carvalho
Reviewer 4 Report
The presented paper is devoted to microalgae-bacteria interaction mechanisms and high value added bioproducts synthesis. I see this review as a well prepared and written. I can recommend this for publication after some changes as follows:
1. Abstract. I recommend to shorten the Abstract before “This review presents…” and to broad the description after “This review presents…”. Expand the annotation of your work. Write about the details of the article.
2. Fig.1 Change “Consumo CO2”
3. Paragraph 2.2 “Oxygen and carbon dioxide ratio” in where?
4. Line 224. I recommend to use “energy industry” or “energy” instead of “thermoelectric”
5. Lines 306-308. I recommend avoiding single sentence paragraph.
6. Conclusion. There is no general summary of the analytical material already done.
7. Conclusion section is also missing some perspective related to the future research work in studied fields.
Author Response
Dear Reviewer,
Thank you for the opportunity to revise our manuscript microorganisms - 1948241 entitled "Obtaining bioproducts from the studies of signals and interactions between microalgae and bacteria". We appreciate the referees´ comments and, considering them, all the alterations were marked up using the “track changes” function in the manuscript. Please, find below the answers to all comments.
Reviewer 4:
The presented paper is devoted to microalgae-bacteria interaction mechanisms and high value added bioproducts synthesis. I see this review as a well prepared and written. I can recommend this for publication after some changes as follows:
- Abstract. I recommend to shorten the Abstract before “This review presents…” and to broad the description after “This review presents…”. Expand the annotation of your work. Write about the details of the article.
ANSWER: Thank you for this note. At the end of the Abstract, we included bottlenecks related to the applications addressed: “...development of new pharmaceutical, cosmetic and food bioproducts from microalgae based on these evaluations, such as prebiotics with effect on intestine microbiota, vegan skincare products, antimicrobial compounds, and culture media with animal free protein for producing vaccines and other biopharmaceutical products…” (lines 18-25 of the revised manuscript).
- Fig.1 Change “Consumo CO2”
ANSWER: Thank you for detecting this mistake. In the revised manuscript, “Consumo CO2” was replaced by “CO2 consumption” (Figure 1, line 132 of the revised manuscript).
- Paragraph 2.2 “Oxygen and carbon dioxide ratio” in where?
ANSWER: Thank you for this observation. In the revised manuscript, “ratio” was replaced by “exchanges” (line 227 of the revised manuscript).
- Line 224. I recommend to use “energy industry” or “energy” instead of “thermoelectric”
ANSWER: Thank you for the suggestion. In the revised manuscript, “thermoelectric” was replaced by “energy industry” (line 238 of the revised manuscript).
- Lines 306-308. I recommend avoiding single sentence paragraph.
ANSWER: Thank you for the suggestion. This sentence was incorporated in the following paragraph (lines 311-316 of the revised manuscript).
- Conclusion. There is no general summary of the analytical material already done.
ANSWER: Thank you for the note. Regarding the conclusion, we have restructured section 5. (lines 634-651 of the revised manuscript).
- Conclusion section is also missing some perspectives related to the future research work in studied fields.
ANSWER: Thank you for this commment. We added future perspectives in the last paragraph of the conclusion (lines 652-658 of the revised manuscript).
At your disposal for any additional necessary information,
Looking forward to hearing from you,
Yours faithfully,
João Carlos Monteiro de Carvalho
Round 2
Reviewer 2 Report
Review. Obtaining bioproducts from the studies of signals and interactions between microalgae and bacteria. Évellin do Espirito Santo, Marina Ishii, Uelinton Manoel Pinto, Marcelo Chuei Matsudo, João Carlos Monteiro de Carvalho.
In this revised version of the manuscript some of the comments rised in the review were included. Still, I consider that most of the antecedent exposed in subsection 4 are not related to the title of the manuscript.
It is my impresion that a clean version of the manuscript could be consider for publication.
Reviewer 3 Report
The authors revised their draft based on the previous comments. I think now this draft has improved and can be published in a journal like Microorganisms.